# Plan2Vec: Unsupervised Representation Learning by Latent Plans

## Abstract

Creating a useful representation of the world takes more than just rote memorization of individual data samples. This is because fundamentally, we use our internal representation to plan and solve problems. In this paper, we introduce **Plan2Vec**, an unsupervised representation learning objective inspired by graph-planning algorithms and value-based reinforcement techniques. By abstracting away low-level control with a learned local metric, we show that it is possible to learn representations that inform long-range structures, entirely passively from high-dimensional sequential datasets that assume no access to action data or an expert policy. Plan2vec learns a latent space through value iteration on the graph formed by the data points, connected via a local metric function trained contrastively from context. We show that the global metric on this learned embedding can be used to plan with $O(1)$ complexity by linear interpolation. This exponential speed-up is critical for planning with a learned representation on any problem containing non-trivial global topology. We demonstrate the effectiveness of Plan2Vec on simulated as well as two real-world image datasets, showing that Plan2Vec can effectively acquire representations that carry long-range structure to accelerate planning. Additional results and videos can be found at https://sites.google.com/view/plan2vec.

## 1 Introduction

Unsupervised representation learning is often motivated by the goal of reducing human involvement in the learning loop, such that an algorithm can learn directly from streams of unlabeled data. Much focus has been placed on an algorithm's transfer performance across supervised learning tasks. However, for control tasks, where an agent is planning on the learned representation space, performance is often poor (Watter et al., 2015). A quick glance at the representation learned by a variational auto-encoder (VAE) reveals that the learned embedding often contains local patches that are quite reasonable by themselves, but the global structure of the learned embedding is often "crumpled", such that Euclidean lines between points that are sufficiently far-apart either cross domain-boundaries, or otherwise detach from the support of the learned manifold (see Fig.4). This observation implies that although the VAE objective encourages embeddings that behave well locally, memorizing individual image samples by reconstruction is insufficient to attain meaningful global structure. This raises an interesting and important question: if rote memorization is insufficient, what else do we need in order to build an agent that can make good plans?

Various works in learning representations for planning attempt to address this issue by learning locally constrained generative models (Banijamali et al., 2017; Watter et al., 2015; Kurutach et al., 2018). One line of work, motivated by controlling complex dynamic systems directly from high-dimensional input, attempts to learn generative models that explicitly impose a reduced local linearity constraint on the learned dynamic manifold (Watter et al., 2015; Banijamali et al., 2017). Such methods contain three major shortcomings. First, some of these formulations rely on learning a forward model, which can not be applied to datasets where action data is unavailable or ill-defined. Second, these generative models rely heavily on the inductive prior within image generation, that images nearby in pixel space are semantically similar, which limits the applicability of these methods to domains

where visual similarities map well to the conceptual space. Finally, the linear constraint and the optimization objective are both local, yet making plans involves non-local concepts of distances and direction. How to learn from streams of observation data to attain a cognitive map of the problem domain without relying on the image similarity priors provided by generative models remains an open problem.

In this work, we pose the problem of unsupervised learning a plannable representation as learning a map of the domain without access to the underlying sampling process and the environment. Such a map has two main properties: First, the map informs conceptual distance between *any* pair of observations, beyond the typical limit of a short spatiotemporal window (Perozzi et al., 2014; Caron et al., 2018). Second, this map has to be consistent with a local metric[1], which is easily attained via self-supervision. Motivated by this problem, we propose **Plan2Vec**, a method for unsupervised representation learning that incorporates planning as part of the learning objective. The technical challenges of this work are threefold: First, the standard formulation of reinforcement learning requires substantial *human supervision* in the form of meticulously *shaped, dense rewards.* Different tasks usually require different reward functions, making it difficult to scale across multiple tasks. The second issue is that reinforcement learning is *active*, as it requires access to an environment between optimization phases to receive trajectories in order to learn. Third, to plan on a continuous state and action space, one usually needs to learn a closed-form behavior *policy* that outputs actions, or a forward model of the environment with actions as an input.

The main contributions of this paper is to overcome all three of these problems by formulating the problem of learning the global structure of a data manifold as a goal-directed reinforcement learning task. To solve the issue of offering a reward, we train a local metric function from local context without supervision, and use it as a sparse reward for reaching the goal with hind-sight relabeling. To address the necessity of active RL and extending RL to a passive setting, we remove the need for either action data, or a model of the world, by planning entirely in the latent configuration space on a graph. Goal-directed planning can be done with a 1-step greedy policy in the learned representation space. We call our method Plan2Vec, for learning an embedding space for planning.

To help illustrate our method, we lead the introduction of Plan2Vec with a set of toy tasks on simulated navigation domains, and show visualization of the components and learned manifold. We then evaluate Plan2Vec under two challenging task settings: First, we show that we can learn representations on deformable objects such as a piece of rope, which is otherwise hard to model. Moreover, we show qualitative results on visual plans between pairs of rope configurations that are randomly selected from the dataset. Second, we tackle real-world navigation on StreetLearn (Mirowski et al., 2018), where we learn to embed a map directly from videos of a car driving through the streets, with *no* access to the ground-truth GPS location data. We show quantitatively that under a constrained planning computation budget, the embedding that Plan2Vec learns using a globally consistent planning objective outperforms baselines that only plan with the local metric.

## 2 RELATED WORKS

The work most similar to ours from the manifold learning community is DeepWalk (Perozzi et al., 2014). DeepWalk aims to embed a social graph by randomly sampling trajectories, then use skip-gram (Mikolov et al., 2013) to embed each graph node contrastively from its contrast. This is related to the contrastive learning objective we use to train our local metric function. Despite of this, the random walk DeepWalk employs to sample those trajectories is limited in terms of distance of travel. As a result it falls under the category of representation learning algorithms that only learn from a localized context. Similarly there is a strong connection between our value-iteration learning objective and the diffusion map literature. In diffusion map, the distance on the learned manifold measures the "diffusion distance" between two points on a graph $\mathcal{G}$ under a markovian transition kernel (Socher & Hein, 2008). One can consider value-iteration as a non-parametric version of diffusion

---

[1]This can be relaxed into a pseudo metric, allowing different images to have zero conceptual distances in-between. This does not affect the applicability of our approach.

map using neural networks for the kernel. A critical difference is that the transition kernel in diffusion maps is not condition on a goal whereas the policy does. As a consequence, the diffusion distance fall-off exponentially as the number of steps increases, just like with DeepWalk. Locally linear embeddings (LLE) could be considered a "stronger" version of skip-gram, where linear contributions of each neighbor is preserved. However, LLE enforces global structure, and prevent volume collapse via addition of a global volume regularization term. This is similar to the variational prior in a variational auto-encoder (VAE) in that both lack meaningful alignment with planning semantics. Recent work in "robust features" point to a connection between the injected noise and the alignment between the input and output manifoldsIlyas et al. (2019) that might be an interesting direction to explore.

Other works look at planning over a graph with various assumptions (Savinov et al., 2018; Zhang et al., 2018; Eysenbach et al., 2019). With semi-parametric topological memory (SPTM) Savinov et al. (2018) the focus is on solving navigation instead of learning a vector embedding. All of these methods use Dijkstra's shortest path first (SPF) search (Dijkstra, 1959) on the graph to plan, resulting in a worst case bound of $O(|E| + |V| \log |V|)$ in computation time where $|E|$ is the number of edges on the graph and $|V|$ the number of vertices. In our experiment, we show quantitatively that the local embedding SPTM uses to make plans is insufficient if the hard-coded planner has a restricted planning budget, whereas the globally consistent representation that Plan2Vec learns via value iteration still plans well. Our result illustrates the importance for an agent to acquire such a global view of the domain and use it as heuristic for planning.

There are also gradient-based planning methods that require expert trajectories, which is a strong assumption that we do not require. Universal Planning Networks and Distributional Planning Networks (Srinivas et al., 2018; Yu et al., 2019) rely on supervised learning to get to the reward through a differentiable forward model, trained end-to-end by grounding through expert actions. Gupta et al. (2019); Tamar et al. (2016) assume that the feature vectors live on a 2D grid world with known environment dynamics and well-defined local connectivity between states.

There have also been Laplacian methods to learn representations efficient for planning (Wu et al., 2019), using a spectral graph objective for approximating the Laplacian eigenfunctions and using L2 distance in this space as a dense reward for goal-directed tasks. This learns a similar type of space that is a compression of the original graph, but assumes the original graph is fully given through acting in the environment as opposed to a disjoint set of trajectories which are connected via a learned local metric.

Embed to control (E2C), RCE, L-SBMP and causal InfoGAN (Watter et al., 2015; Banijamali et al., 2017; Ichter & Pavone, 2018; Kurutach et al., 2018) are a line of generative model that explicitly incorporate forward modeling in the latent space. They show that the learned representation is *plannable*, without directly incorporating a planner as part of their learning objective. Our goal is drastically different – Plan2Vec learns a representation *by planning*, as opposed to just showing *one can plan with a learned representation*. Plan2Vec explicitly acquires the concept of "reachability" conditioned on an optimal policy as part of the representation. This results in a semantically meaningful and locally consistent global structure.

Another branch of work coming from the reinforcement learning community are self-supervised or task-agnostic RL (Florensa et al., 2019; Kahn et al., 2017; Pong et al., 2019). These work aim to reduce the amount of human involvement in designing reinforcement learning algorithms for individual tasks. Plan2Vec is distinguished from these proposals in that we do not aim to learn a policy distribution $\pi(a|o)$. Instead, we want to learn generalizable representations of the environments that makes learning such a low-level policy, or running classical control algorithms more efficient. By abstracting away the actions, Plan2Vec is able to plan over much longer horizons, as demonstrated in Sec. 5.

## 3  Technical Background

We now overview methods that learn a local metric between pairs of images that are close-by, and proximal dynamic programming under a standard Markov decision process (MDP) formalism (Sutton & Barto, 1998).

**Learning a Local Metric.**  Intuitively, a metric is a bivariate function that gives a measure of similarity between two points. Formally, $f_{a,b\sim D} : (a, b) \rightarrow \mathbb{R}^+$ is a symmetric, real-valued, and positive-definite function over its domain $D \times D$. When distance labels are available one can learn such a function via supervised learning. In reality, however, we often need to work with sequential datasets without access to a sampling policy that is jointly optimized, in which case one cannot assume long-horizon optimality in the sequences we want to learn from. As a result, the distance information between frames of observations is only good up to a limited temporal window, beyond which noise dominates.

In language modeling and unsupervised representation learning domains, it is often easy to construct positive and negative examples, and pose a binary classification objective as a Noise-Contrastive Density Estimator (NCE) (Gutmann & Hyvärinen, 2010),

$$\mathcal{L}_{\text{NCE}} = -\log \frac{f(x_i, c)}{\sum_{x \sim X} f(x, c)} \,, \tag{1}$$

where $f$ is a convex function proportional to the density $p(x, c)$. Minimizing the NCE loss can be mapped to maximizing a lower-bound on the mutual information between the latent code $c$ and the data distribution $X$ (Oord et al., 2018; Hjelm et al., 2018),

$$\mathbf{I}(X, c) = \mathbb{E}\left[\log \frac{P(X|C)}{P(X)}\right] \geq log(N) - \mathcal{L}_{\text{NCE}} \,. \tag{2}$$

Rather than directly learning a representation this way (Sermanet et al., 2017), Plan2Vec extends the standard binary NCE objective to learn a local metric function, and uses it as a reward function.

**Universal Value Function Approximator as a Metric.**  We formulate value iteration under the Markov decision process (MDP) formalism (Bellman, 1957). The MDP is parameterized by the tuple $\langle S, A, P, r \rangle$. $S$ and $A$ are the sets of states and actions, $P(s'|s, a)$ is the transition model of the environment, and $r(s, a, s')$ is the reward function. An agent is represented by its policy distribution $\pi(a|s)$. The state value function $V_\pi : S \rightarrow \mathbb{R}$ represents the expected sum of discounted future rewards for being at state $s$, conditioned on the reward $r$ and the policy $\pi$. In sample-based value iteration with neural networks, we can learn the value function by minimizing the empirical Bellman-residual

$$\delta = \left\| V(s; \theta) - \mathcal{B}_\pi^* V \right\|, \tag{3}$$

where the Bellman optimality operator is defined as

$$\mathcal{B}_\pi^* V = \max_{a_t} \left[ R(s_t, a_t, s_{t+1}) + \gamma \sum_{s_{t+1}} P(s_{t+1}|s_t, a_t) V(s_{t+1}; \theta) \right]. \tag{4}$$

Universal Value Function Approximators (UVFAs) (Schaul et al., 2015) extend this task-specific reward to learn a "universal" value function by generalizing to all goals $g \in S$. The reward now conditions on the goal $r(s, a, s', g)$. Assuming that the goals are uniformly sampled from $S$ and the value function is symmetric, UVFA becomes a metric on $S$ up to a correction constant. If we further assume that the MDP is deterministic, the sample-based Bellman residual can be reduced to

$$V(s, g; \theta) \leftarrow r(s, a, s', g) + \gamma V(s', g; \theta), \tag{5}$$

which we use to learn our latent space, as detailed in Sec. 4.2.

## 4  Learning Representations by Latent Planning

Our goal is to learn a representation that goes beyond rote memorization of the dataset, which consists of disjoint temporal sequences. Critically, we want the structure of the embedding to capture the global topology of the dataset, such that for any observation $o$ in the domain, we can make useful inference with respect to another sample $o_{\text{goal}}$, no matter how far away $o_{\text{goal}}$ is. Having access to such a global metric, $\forall o, o_{\text{goal}}$ pairs, would enable effective planning on non-trivial, high-dimensional, and/or complex topologies that are otherwise prohibitively slow. In this section we give a high-level overview of Plan2Vec, with implementation details available in Appendix **??**. We first define local connectivity between states through a local metric function trained in a self-supervised manner with a contrastive loss. Different from Watter et al. (2015); Banijamali et al. (2017) and similar to Kurutach et al. (2018), our method does not rely on dynamics of the underlying environment in the form of sampled action data, and neither do we learned a forward model, which distracts from long-range planning. Instead,

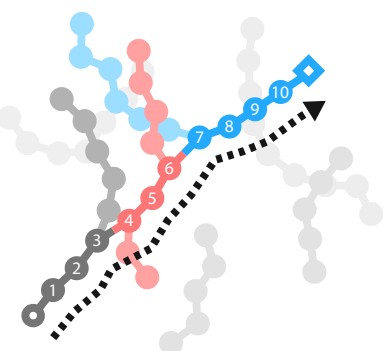

Figure 1: Example of a path (black dashed arrow) found across independent trajectories (colored lines) from an initial state (gray circle) to a goal state (blue square), with learned local metric creating new connections.

this task occurs on a graph where disjoint temporal sequences are connected by the local metric function. We learn a representation space that encodes paths on the graph such that a 1-step greedy policy on the learned representation is equivalent to a path finding algorithm on the graph.

### 4.1 Noise-Contrastive Learning the Local Metric

In many representation learning problems, one has access only to noisy binary or categorical learning signals. It is often easy to find symmetry transformations in a particular problem that make it trivial to define a binary or ordered categorical relationship between data-points. In skip-grams (Mikolov et al., 2013; Jozefowicz et al., 2016) the classifier decides

---

**Algorithm 1** Contrastive Local Metric Learning

**Require:** set of observation sequences $\{\tau = x_{[0:T]}\}$
1: Initialize $f_\phi$
2: Sample $x_t$ and $x_{\mathbf{I}} = x_t$, $y^{\mathbf{I}} = 0$
3: Sample $x_t, x_{t+1}^+$ where $x_t, x_{t+1} \in \tau_i$, $y^+ = 1$
4: Sample $x_t, x^-$ where $x^- \sim \tau_j$, $x_t \notin \tau_j$, $y^- = 2$
5: **for** each epoch **do**
6: $\quad$ minimize $\|f_\phi(x, x^*) - y^*\|_2$ for $x, x^{\pm,\mathbf{I}}, y^{\pm,\mathbf{I}}$
7: **end for**

---

whether a word belongs to a certain context. In time-contrastive networks (Sermanet et al., 2017) classifiers decide whether two views correspond to the same scene. In our case, we extend this dichotomy to one of {*identical*, *close*, or *far-apart*}. Formally this can be considered as a natural extension of the standard definition of a metric from the positive real-line to a directed set where each element in the set corresponds to one of the categories. To reflect the order between the category labels, we use a regression objective. The labels are designated 0 for identity, $k/K$ for true neighbors that are k steps apart if $k \leq K$, and 2 for negative samples for other trajectory or the same trajectory but more thank $K$ steps apart. Alg. 1 explains the procedure in detail. Fig. 3 illustrates the well-behaved distribution of local distance scores for one of our experimental domains. Visualization of pairs show new transitions that are not present in the training trajectories.

### 4.2 Extrapolating Local Metric to A Globally Consistent Embedding By Planning

To extrapolate the local metric information to a globally consistent embedding that can speed up planning, we first connect those disjoint trajectories in the dataset using the new connections found by the trained local metric function $f_\phi$ (see Alg. 1 and Fig. 3c). One can use this graph defined by $f_\phi$ to perform planning at inference time with a path finding algorithm. However, this takes $O(|E| + |V| \log |V|)$ time (Fredman & Tarjan, 1987) where $|E|$ is the number of edges and $|V|$ is the number of vertices in the graph. This quickly becomes intractable for large graphs.

Instead, we can move this computation to training time to achieve fast inference by learning a representation that distills shortest path information from this graph. Our goal is to learn an embedding on which there exists a metric that correctly reflects the difference in reachability between points in the neighborhood of the current observation, and the goal. Now formulated as an reinforcement learning problem, this is equivalent to learning a goal-conditioned value function $V_\Phi(s, g) := \|\Phi(s) - \Phi(g)\|_p$ at state $s$ towards the goal $g$, where in practice we take $p = 2$. Hence, the value function is defined as the Euclidean distance between the two states in the learned embedding space.

---

**Algorithm 2** Unsupervised Learning by Latent Plans

**Require:** planning horizon $H$
**Require:** set of observation sequences $S = \{\tau = x_{[0:T]}\}$
**Require:** local metric function $f_\phi(x, x') \Rightarrow \mathbb{R}^+$
**Require:** reward function $r(x, x_g) = -f_\phi(x, x_g)$
1: Initialize global embedding $\Phi(x) \Rightarrow \mathbb{R}^+$
2: **repeat**
3:     sample $x_0, x_g \in S$ as *start* and *goal*
4:     **repeat** {h=0, h++}
5:         find *set* $\mathbf{n} = \{x'$ s.t. $f_\phi(x_0, x') \in N(1, \epsilon)\}$
6:         find $x^* = \arg\min_{x \in \mathbf{n}} f_\phi(x, x_g)$
7:         compute $r_t = r(x^*, x_g)$
8:         add $\langle x, x^*, r_t, x_g \rangle$ to buffer $B$
9:     **until** r = 0 or h = $H$
10:    Sample $\langle x, x', r, x_g \rangle$ from $B$
11:    minimize $\delta = \left\| V_\Phi(x, x_g), r + V_\Phi(x', x_g) \right\|_p$ where $V_\Phi(x, x_g) := \|\Phi(x) - \Phi(x_g)\|_p$
12: **until** convergence

---

Similarly, the local metric $f$ becomes the *cost* to travel the distance between state $s$ and the next step $s'$. The action set $\mathcal{A}(s)$ for the agent consists of a flexible number 1-step neighbors, sourced from the local metric function $f_\phi$ for each node $s$ in the graph. As a reminder, $f_\phi$ is a regression function that outputs a score designating "closeness" between a pair of states $x, x'$. In practice, we select a subset $x' \in X$ where $f_\phi(x, x') \in N(1, \epsilon) := [1 - \epsilon, 1 + \epsilon]$ with tuned hyperparameter $\epsilon$ and $X$ is the set of all states from the dataset. Choosing small $\epsilon$ leads to shorter steps between states and longer paths, and vice versa. We have now created a reinforcement learning task and can learn this value function with multi-step value iteration using transitions sampled from the graph (see Alg. 2). To improve rate of learning, we use hindsight experience re-labeling (Andrychowicz et al., 2017) to insert positive reaching examples.

## 5 EXPERIMENTAL EVALUATION

In this section, we experimentally answer the following questions: 1) Can we build a graph from sequential datasets using a contrastively trained local metric? 2) Can we extrapolate this local metric to a global embedding, and make planning easier? 3) Would Plan2Vec work in domains other than navigation, and learn features that are not visually apparent? To answer these questions, we show quantitative results on simulated 2D navigation. Then we extend Plan2Vec to the challenging deformable object manipulation tasks. Finally, we show that Plan2Vec can learn non-visual features of the domain where other methods perform poorly, on a real-world large-scale street view dataset.

### 5.1 SIMULATED NAVIGATION

Our first domain is a room with a continuous, 2-dimensional state space. A camera looks down on a square arena with a robot (blue block). The trajectory data consist of top-down images of the arena. We use ground-truth coordinates for evaluation only. Our experiment covers three room layouts with increasing level of difficulty: an open room, a room with a table in the middle, and a room with a wall separating it into two corridors that resembles a C-shaped maze (see Fig. 2).

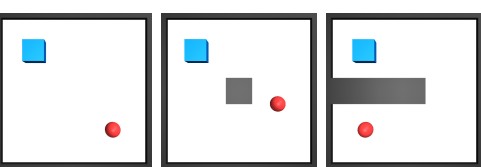

Figure 2: Simulated 2D navigation environments. We use these domains to illustrate the various properties of our method. *left*: Open, *middle*: Table, *right*: C-Maze. The blue block is the agent, and the red circle indicates the desired goal.

**Connecting The Dots by Generalization.** We first investigate if the contrastively trained local metric function generalizes. To train the local metric contrastively on this

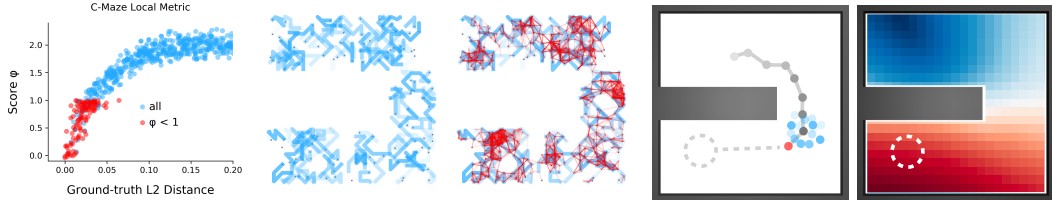

Figure 3: (1) Local metric score in comparison to ground-truth $L_2$ distance with predicted neighbors in red. (2) Trajectories given in the dataset. (3) Points from different trajectories are connected by generalizing the local-metric function. Out-of-training-set Connections shown in red. (4) Step sequence in C-Maze, learned via Plan2Vec. Gray dashed circle is the goal position. Red dot is the planned next step (1-step), greedy w.r.t the global metric function being learned. Blue dots are the neighbors sampled using the local metric function. Gray dot indicates the current and past positions of the agent. Sequence shows the agent getting around the wall in C-Maze. (5) Learned value function for a goal location on the bottom left corner (white dashed circle). Blue color is further away, red is close.

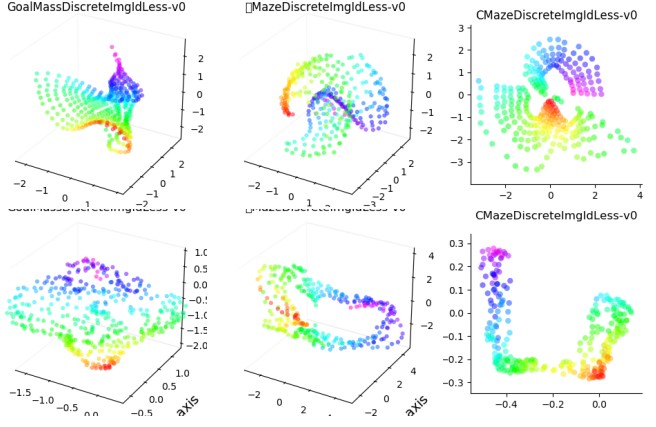

Figure 4: Learned Embedding with VAE (top row) vs Plan2Vec (bottom row). The columns correspond to the *Open Room*, *Table*, and *C-Maze* domains. Representation learned by the VAE is wrapped globally. Whereas Plan2Vec learns a globally coherent embedding. In C-Maze, the two ends of the tunnel are further apart, correctly reflecting the decrease in reachability between those points.

Figure 5: (left and middle) Difference in learned global metric on Open Room and C-Maze. The goal used to query the value map is indicated by the dashed circle. (right) shows the agent getting around the wall with the learned embedding (blue), whereas a Euclidean planner gets stuck.

domain, we restrict $K = 1$, such that only observations that are 1-step away are considered neighbors. The local metric predicts a distance score that is between 0 and 2, where 0 corresponds to identical observations, 1 to neighbors, and 2 to observations that are further apart. Fig. 3a shows the distribution of the score against ground-truth distance. In short ranges, the learned model is able to recover the local metric but saturates as distances increase. The score is well-behaved enough that it is easy to pick suitable values for the neighbor threshold (indicated by the ceiling of the red points). We plot new transitions found by the local metric against those in the dataset (blue). Fig. 3b visualizes the sampled trajectories (in blue, of length 4), whereas Fig. 3c shows the new ones found by the learned local metric function.

**Accelerating Planning with a Learned Cognitive Map.** To evaluate Plan2vec, we compare with SPTM (Savinov et al., 2018) and VAE (Kingma & Welling, 2013) learned representations under a restricted planning budget. Table 1 shows the success rates on the 2D navigation domains when the planning horizon is limited to a single step in the future. Under this regime, SPTM

Table 1: Planning Performance on 2D Navigation

| State Input | Open Room | Table | C-Maze |
|---|---|---|---|
| Euclidean | $100.0 \pm 0.0$ | $96.3 \pm 1.4$ | $88.7 \pm 3.6$ |
| Plan2vec (L2) | $100.0 \pm 0.0$ | $96.6 \pm 0.9$ | $86.0 \pm 4.1$ |
| Plan2vec (pseudo) | $96.9 \pm 0.5$ | $96.7 \pm 2.0$ | $83.1 \pm 3.0$ |
| **Image Input** | | | |
| Plan2vec (L2) | $\mathbf{90.0 \pm 2.0}$ | $\mathbf{76.4 \pm 9.2}$ | $\mathbf{80.2 \pm 6.3}$ |
| SPTM (1-step) | $39.7 \pm 6.1$ | $23.7 \pm 6.1$ | $31.4 \pm 6.5$ |
| VAE | $73.9 \pm 4.3$ | $30.2 \pm 6.5$ | $52.7 \pm 5.8$ |
| Random | $3.2 \pm 2.5$ | $3.5 \pm 2.5$ | $4.7 \pm 2.8$ |

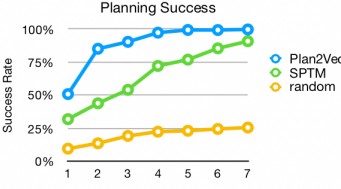 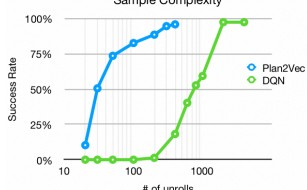 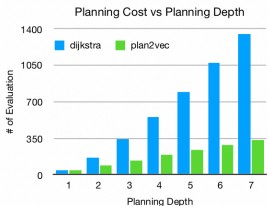

Figure 6: Left: Comparison of Plan2Vec with full SPTM and a random baseline for varying plan lengths. Center: Comparison of Plan2Vec with DQN in terms of sample efficiency. Right: Comparison of computation time of Plan2Vec and Dijkstra's.

fails to succeed most of the time. This is because the local similarity function used in the parametric memory does not contain long-range information about the domain, and hence is insufficient as a planning heuristic for a memoryless planner. The VAE learned embedding does better on the Open Room domain, but falls short on more complex room arrangements. In comparison, the representation learned by Plan2Vec succeeds most of the time. To investigate this further, we visualize the learned global embedding for VAE versus that of Plan2Vec (Fig. 4). With the Open Room, Plan2Vec learns a latent space that looks flat. With Table and C-Maze, two points that are close in Euclidean space but separated by the wall appear far away in the learned latent space, reflecting the reduced reachability in between. For latent space higher dimension than 3, we can directly visualize the value function as shown in Fig. 5.

We also include comparison to full SPTM, and find that Plan2Vec performs better at varying planning lengths, compare sample complexity with DQN (Mnih et al., 2015) using the local metric as a reward function, and computation time for planning of Dijkstra's and Plan2Vec to empirically validate the asymptotic bounds given in Section 4.2, all shown in Figure 6. These are all run on state space of the Open Room domain.

## 5.2 Manipulation of Deformable Objects

While we have made strides in controlling rigid bodies with reinforcement learning, manipulating deformable objects still remains an open problem. Methods so far rely on learning a generative model over the image sample (Kurutach et al., 2018). To learn a plannable representation in a purely discriminative manner we now apply our method to the rope dataset (Wang et al., 2018). The rope dataset is composed of 18 independent trajectories with 14k images total. Each image is a gray scale photo of a piece of rope wrapped around two pegs that are fixed on the table surface. The two pegs help define distinct topology for the configuration of the rope that needs to be respected for reasonable transitions. The challenge with the rope dataset is that it does not have a well-defined low-dimensional configuration space, making it difficult to design quantitative evaluation metrics. To get around this issue, we evaluate our method with planning on single trajectories, where the original sequences of observation can be used as qualitative baselines. We do find that our local metric generates a connected graph over all 18 trajectories, therefore there exists a viable plan from *any* image $o_s$ to *any* goal image $o_g$. The difficulty of the planning problem varies with the connectivity of the graph, which is in turn dictated by the threshold set on local metric $\phi$.

Fig. 8 shows the distribution of neighbors for a set threshold $T = 1.1$, with both in and out-trajectory neighbors. This highlights the difficulty of the rope manipulation task and learning a latent representation that reflects a sparse connectivity graph. Fig. 9 shows an example of a plan generated by Plan2Vec for a given start and goal state, where we can see that each transition only perturbs the configuration of the rope locally.

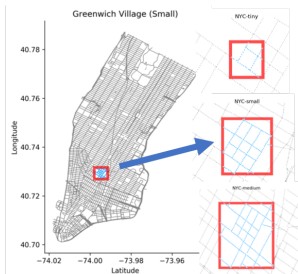

Figure 7: A visualization of the three datasets Tiny, Small, and Medium used in our experiments.

## 5.3 Beyond Visual Similarity: Real-world Navigation

To answer the question of whether Plan2Vec is able to learn non-visual features of the domain, we evaluate on a visual navigation task using the real world dataset StreetLearn (Mirowski et al., 2018). In comparison to the previous two tasks, the StreetLearn dataset offers an interesting alternative because the spatial re-

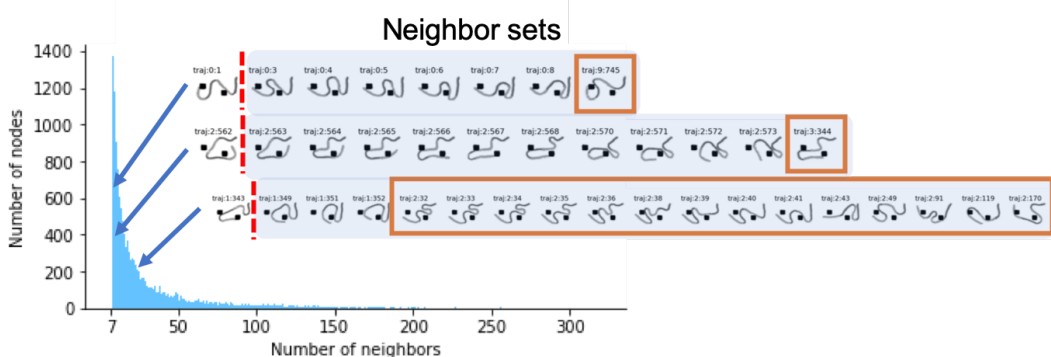

Figure 8: Histogram of number of neighbors per node, with visualized examples. Original nodes are left of the red dashed line and out of trajectory neighbors are outlined in orange.

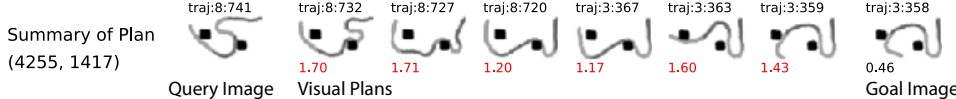

Figure 9: Example of visual plan generated by Plan2Vec on the Rope Domain showing steps coming from two different trajectories (8 and 3). Each transition only perturbs the configuration of the rope locally. The numbers above denote the trajectory and time step the image is from, the number below represents the score by the local metric $f_\phi$.

lationship between views at different locations is not visually apparent. One can not easily tell that Union Square is to the north of Washington Square Park from street views alone. Yet a city resident knows exactly which general direction to turn to. This is in stark contrast to both the Room domain and the rope domain, where visual similarity is easily mapped to being close in the configuration space. We quantitatively evaluate the planning performance of Plan2Vec versus the VAE baseline in Table 2 using generated datasets (Fig. 7). This result shows that the planning performance of VAE on the StreetLearn dataset is barely above that of a random baseline. This is a common short-coming with unsupervised models that rely on the inductive prior of the generator to learn. Plan2Vec, on the other hand, uses planning as a general framework to extend any type of local and semantically meaningful signal to a consistent global embedding. We interpret these results by hypothesizing that Plan2Vec is successful in learning *non-visual concepts of reachability* (in this case an idea of the map), whereas VAE only clusters the images by visual similarity.

In Table 2, we also include comparisons with SPTM, where the agent is only allowed to plan 1-step ahead. This computation-constrained regime is interesting because a good planning heuristic is critical for good search performance. The result shows that in this regime, Plan2Vec performs well above SPTM, which backs our intuition that a good representation can and should alleviate some of the computational cost of planning at test time.

Table 2: *1-step* Planning Performance on StreetLearn. Numbers are percentage of success for reaching goals that are within 50 steps of the starting point. Full Graph Search methods succeed 100% of the time.

| | Success Rate (%) | | |
| --- | --- | --- | --- |
| **StreetLearn** | Tiny | Small | Medium |
| Plan2vec (Ours) | **92.2 ± 2.9** | **57.2 ± 4.3** | **51.4 ± 6.9** |
| SPTM (1-step) | 31.5 ± 5.8 | 19.3 ± 5.8 | 20.2 ± 5.2 |
| VAE | 25.5 ± 5.6 | 14.4 ± 4.8 | 16.9 ± 5.5 |
| Random | 19.9 ± 5.4 | 12.0 ± 5.2 | 12.7 ± 4.6 |

Formally, Plan2Vec's 1-step greedy planning is $\mathcal{O}(1)$ at test time, whereas SPTM is $\mathcal{O}(E)$ where $E$ is the size of the graph. This also shows that Plan2Vec memorizes information that is computationally more valuable. Lastly, we observe that Plan2Vec generalizes – despite the agent never having seen a particular combination of starts and goals in the original dataset – by successfully navigating using the values acquired during training time as evidenced by the large jump in performance compared to other methods in Table 2.

## 6  CONCLUSION

We have presented an approach to attain a globally consistent representation from streams of observation data in a purely unsupervised fashion without generating images. Integral to our approach is the incorporation of planning as part of our learning objective, to enforce the semantic notion of reachability between any pair of images on the learned embedding. This differs our approach from previous work in learning *plannable representations* – in that the plannability is a consequence of the planning objective, instead of local linearity constrains. In addition, we realize that formulating unsupervised learning as a reinforcement learning problem has the added benefit of allowing one to insert arbitrary local information about the domain as the reward $R(s, s')$, and the explicit including of a maximizing inner step.

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

## Appendix

## A    Implementation Details

### A.1    Architecture and Parameter Tuning

In our experience, we found that it is substantially easier to learn the local metric function, versus the global metric. This is reflected in the network architecture that is needed for the network to converge. For the local metric function, we typically achieve high-90% testing accuracy on the fixed dataset. For the global metric function, we conduct architecture search with supervised training. Results are averaged over 3 seeds that are not hand-picked.

### A.1.1    Simulated Navigation with Proprioceptive Inputs

The local metric function is a simple 4-layer perceptron. To learn the local metric, we run 20 environment simulation in parallel, collect 50 rollouts each (1000 in total). Each rollout has 10 timesteps. We then generate data pairs that are 1-step apart. We use a mini batch size of 32, and a learning rate of $3 \times 10^{-4}$. The rest of the Adam optimizer hyper parameters are stock ones from pyTorch.

```
LocalMetric(
  (embed): Linear(in_features=2, out_features=50, bias=True)
  (head): Sequential(
    (0): Linear(in_features=100, out_features=50, bias=True)
    (1): ReLU()
    (2): Linear(in_features=50, out_features=50, bias=True)
    (3): ReLU()
    (4): Linear(in_features=50, out_features=50, bias=True)
    (5): ReLU()
    (6): Linear(in_features=50, out_features=1, bias=True)
  )
)
```

In order for the global metric function to converge on the same dataset, it needs to be wider than the local metric function. This tangentially shows that the global metric is learning more complex information than the local metric.

```
GlobalMetricDistance(
  (embed): Sequential(
    (0): Linear(in_features=2, out_features=200, bias=True)
    (1): ReLU()
    (2): Linear(in_features=200, out_features=200, bias=True)
    (3): ReLU()
    (4): Linear(in_features=200, out_features=200, bias=True)
    (5): ReLU()
    (6): Linear(in_features=200, out_features=100, bias=True)
    (7): ReLU()
    (8): Linear(in_features=100, out_features=2, bias=True)
  )
  (head): (self.head = (lambda a, b: (a - b).norm(2, dim=-1))
  )
)
```

### A.1.2    Network Architecture used in simulated navigation with image inputs

The local metric function is a stacked five-layer convolution network. To make learning easier, we stack the two input images channel wise, so that there is maximum flexibility for the local metric function.

```
LocalMetricConvLarge(
  (trunk): Sequential(
    (0): Conv2d(2, 32, kernel_size=(4, 4), stride=(2, 2))
    (1): BatchNorm2d(32, eps=1e-05, momentum=0.1, affine=True)
    (2): ReLU()
    (3): Conv2d(32, 64, kernel_size=(4, 4), stride=(2, 2))
    (4): BatchNorm2d(64, eps=1e-05, momentum=0.1, affine=True)
    (5): ReLU()
    (6): Conv2d(64, 64, kernel_size=(4, 4), stride=(2, 2))
    (7): BatchNorm2d(64, eps=1e-05, momentum=0.1, affine=True)
    (8): ReLU()
    (9): Conv2d(64, 32, kernel_size=(4, 4), stride=(2, 2))
    (10): BatchNorm2d(32, eps=1e-05, momentum=0.1, affine=True)
    (11): ReLU()
    (12): View(-1, *(128,))
    (13): Linear(in_features=128, out_features=128, bias=True)
    (14): ReLU()
    (15): Linear(in_features=128, out_features=100, bias=True)
    (16): ReLU()
    (17): Linear(in_features=100, out_features=1, bias=True)
  )
)
```

We increase the capacity of the network for the global metric.

```
GlobalMetricConvL2_s1(
  (embed): Sequential(
    (0): Conv2d(1, 128, kernel_size=(7, 7), stride=(1, 1))
    (1): BatchNorm2d(128, eps=1e-05, momentum=0.1, affine=True)
    (2): ReLU()
    (3): Conv2d(128, 256, kernel_size=(7, 7), stride=(1, 1))
    (4): BatchNorm2d(256, eps=1e-05, momentum=0.1, affine=True)
    (5): ReLU()
    (6): Conv2d(256, 256, kernel_size=(7, 7), stride=(2, 2))
    (7): BatchNorm2d(256, eps=1e-05, momentum=0.1, affine=True)
    (8): ReLU()
    (9): Conv2d(256, 256, kernel_size=(7, 7), stride=(2, 2))
    (10): BatchNorm2d(256, eps=1e-05, momentum=0.1, affine=True)
    (11): ReLU()
    (12): Conv2d(256, 256, kernel_size=(7, 7), stride=(2, 2))
    (13): BatchNorm2d(256, eps=1e-05, momentum=0.1, affine=True)
    (14): ReLU()
    (15): Conv2d(256, 256, kernel_size=(2, 2), stride=(1, 1))
    (16): ReLU()
    (17): View(-1, *(256,))
    (18): Linear(in_features=256, out_features=2, bias=True)
  )
  (head): (self.head = (lambda a, b: (a - b).norm(2, dim=-1))
  )
)
```

### A.1.3   NETWORK ARCHITECTURE USED IN ROPE AND STREETLEARN

To obtain good performance on the vision task, we found that is it helpful to introduce the ResNet18 architectureHe et al. (2016). We verify the learning capacity of this architecture by running supervised learning on the StreetLearn dataset. For the local metric function, a regular convolution network that is deeper and wider is sufficient.

```
LocalMetricConvDeep(
```

```
  (trunk): Sequential(
    (0): Conv2d(2, 128, kernel_size=(4, 4), stride=(2, 2))
    (1): BatchNorm2d(128, eps=1e-05, momentum=0.1, affine=True)
    (2): ReLU()
    (3): Conv2d(128, 128, kernel_size=(4, 4), stride=(2, 2))
    (4): BatchNorm2d(128, eps=1e-05, momentum=0.1, affine=True)
    (5): ReLU()
    (6): Conv2d(128, 128, kernel_size=(4, 4), stride=(1, 1))
    (7): BatchNorm2d(128, eps=1e-05, momentum=0.1, affine=True)
    (8): ReLU()
    (9): Conv2d(128, 128, kernel_size=(4, 4), stride=(1, 1))
    (10): BatchNorm2d(128, eps=1e-05, momentum=0.1, affine=True)
    (11): ReLU()
    (12): Conv2d(128, 128, kernel_size=(4, 4), stride=(1, 1))
    (13): BatchNorm2d(128, eps=1e-05, momentum=0.1, affine=True)
    (14): ReLU()
    (15): Conv2d(128, 128, kernel_size=(4, 4), stride=(1, 1))
    (16): BatchNorm2d(128, eps=1e-05, momentum=0.1, affine=True)
    (17): ReLU()
    (18): View(-1, *(512,))
    (19): Linear(in_features=512, out_features=128, bias=True)
    (20): ReLU()
    (21): Linear(in_features=128, out_features=100, bias=True)
    (22): ReLU()
    (23): Linear(in_features=100, out_features=1, bias=True)
  )
)
```

The global metric function uses a ResNet18 trunk.

```
ResNet18L1(
  (embed): ResNet18(
    (conv_1):
        nn.Conv2d(1, 64, kernel_size=7, stride=2, padding=3, bias=False)
    (ResNet18): ResNet18([2, 2, 2, 2])
  )
  (head): (self.head = (lambda a, b: (a - b).norm(p=1, dim=-1))
  )
)
```

