# OpenReview forum: "Plan2Vec: Unsupervised Representation Learning by Latent Plans"
_ICLR.cc/2020/Conference — Reject_

### Official Review · AnonReviewer3 · 2019-10-23
**Official Blind Review #3**

**Rating:** 1

**Review:**

The paper proposes to learn a representation space that is useful for planning. This is done by a 2-step process: (1) learn a metric to reflect locality, (2) utilize that metric to learn a global metric by utilizing UVFAs from Reinforcement Learning literature. While the idea may be an interesting direction to improve sample efficiency of planning, it is not the first work that proposes to combine planning with representation learning, and I do not think the work is clearly presented/motivated/validated sufficiently to deem it acceptable.

As presented, there are many details that are unclear or imprecise. I list them below.

Feedback:
(1) The contrast between learning playable representations vs. planning via representation learning (as proposed) is  poorly motivated. The presentation ignores well recognized works in the latter in terms of UPNs [1], VINs [2].
(2) The RL background section is poorly presented -- as written Equation (4) seems incorrect. Further, is a dynamics model learned to be able to utilize the correct form of Equation (4)? It is never specified. How exactly is a plan extracted for any of the experiments?
(3) Possibly wrong citation for Equation (2) -- maybe cite the Oord et. al. paper? Further, c vs C -- the notation in general in the paper is inconsistent and poor.
(4) Algorithm 1, line 4 -- typos for x. Further for Contrastive learning methods to be effective, the negative sampling needs to be much higher. What ratio of samples is used for local metric learning?
(5) If an actual forward dynamics model is not used (Section 3 end), then as given in Algorithm 2 from pseudocode I'm completely unsure how a plan is extracted -- How is the UVFA used to extract a plan? What is N(1,\epsilon)? P-norm?
(6) The Dijkstra psuedocode is rather incomplete.
(7) How is sampling done with the local metric function? How is a plan generated? These important details are missing.
(8) Figure 4 is poorly presented/annotated. In the description "Plan2Vec correctly stretched out learned.." -- no it doesn't, visually it seems wrapped too.
(9) There exists literature in RL to combine the 2 step metric learning process to 1 step. This is relevant. [3].
(10) What is the action space of these domains? Based on visualization in Figure 3 (2), are the actions continuous? What is the action space for Figure 7? These details details are missing.
(11) Description of the StreetLearn dataset would be useful. Further an example of why Plan2Vec generalizes (last para, Section 4.3) would be useful. Just statement based claims seem rather vacuous.
(12) The last statement in Conclusion - why? The paper has made an argument against utilizing generative modelling in an unsupervised manner. So why would including it improve it? Such unexplained statements reflect poorly.

Questions:
(1) What do you see as the contribution of the work? Why is it new/different from existing literature?
(2) How exactly do you generate a plan?
(3) How is the UVFA V used?
(4) When to use UVFA? When to use Dijkstra? Why are the choices in the experiments as made?


Some typos to help future version of the paper:
(1) Section 2 -- We now overview --> We now review.
(2) Section 2, UVFAs -- expected discounted future value --> expected sum of discounted future rewards.
(3) Section 2 completely ignores the discount factor/horizon of an MDP, although the utility here I suppose relies on the horizon aspect.
(4) Figure 6 explanation is very sloppy (description and body).
(5) The Zhang et. al. reference in Section 4.2 is unclear.

While I am not from the planning community, I am from the RL community - and as presented the paper is ignoring a lot of details, and was extremely difficult to piece together for me.

[1] Srinivas, Aravind, et al. "Universal planning networks." arXiv preprint arXiv:1804.00645 (2018).
[2] Tamar, Aviv, et al. "Value iteration networks." Advances in Neural Information Processing Systems. 2016.
[3] Wu, Yifan, George Tucker, and Ofir Nachum. "The laplacian in rl: Learning representations with efficient approximations." arXiv preprint arXiv:1810.04586 (2018).


**Experience Assessment:**

I have read many papers in this area.

**Review Assessment: Checking Correctness Of Derivations And Theory:**

I carefully checked the derivations and theory.

**Review Assessment: Checking Correctness Of Experiments:**

I carefully checked the experiments.

**Review Assessment: Thoroughness In Paper Reading:**

I read the paper at least twice and used my best judgement in assessing the paper.

---

> ### Author Response · Authors · 2019-11-15
> **Thank you for your thoughtful comment!**
>
>
> We have incorporated additional discussion with respect to gradient-based planning methods such as UPN and VIN.
> Thank you for the catch on Eq 4, it has been updated with the transition probability moved inside to act on the value estimate for next state.
>
> Included the Oord et al. paper citation. A dynamics model is not learned, the local metric is used to source neighbors from the dataset.
>
> On navigation we used two times more negative samples than close neighbors. We did not see issues with performance when visually inspecting neighbors and checking error on a validation set. On the rope dataset, we adjust the ratio is larger.
>
> We replace a dynamics model with the local metric sourcing next step neighbors. We build a lookup table of neighbors by passing all state pairs through the local metric and use a threshold to control number of neighbors.
>  We have removed the Dijkstra’s pseudocode to clarify focus on value iteration to learn a representation as our main method.
>
> We have updated the methods section to clarify how neighbors are sourced using the local metric, and that plans are generated with a greedy policy over neighbors by taking the minimum over Euclidean distances from neighbors to the goal.
>
> We’ve clarified the description of Figure 4. There are local rotations seen, but globally the representation is consistent.
> Thank you for the reference, we’ve added discussion on Laplacian methods [3] to related work.
>
> The action spaces for these environments are all discrete for collecting the datasets except rope domain, but because the action space is abstracted away in our method, the action space can be discrete or continuous. These details are missing because they are not needed by our method, we only assume that a set of disjoint trajectories are given.
>
> The StreetLearn dataset is generated from Google StreetView and consists of 3D photos from pedestrian level of the entire island of Manhattan. From this, we curated several independent randomly generated trajectories over several blocks and perform goal-based navigation from streetview images. There are a limited number of start and goal states in this dataset, and therefore almost all the start and goal states Plan2Vec is evaluated on are never seen before.
>
> We have removed the final sentence in the conclusion for succinctness.
>
> Questions:
>
> (1) What do you see as the contribution of the work? Why is it new/different from existing literature?
>
> In this work, we combine two ideas. The first is to train a local metric with a contrastive loss, and the second is to distill the graph defined by that local metric into a representation space that can perform O(1) planning per step.
> This is novel in that we can frame this in a self-supervised, passive setting with no assumptions made about the action space and without requiring actions in the dataset. While there has been similar work to leverage graph building, it either uses Dijkstra’s at inference time, leading to O(|E|+|V|log|V|) total planning time, or builds the graph using the replay buffer and compresses the graph with Laplacian methods, which does not apply to passive datasets.
>
> (2) How exactly do you generate a plan?
>
> At training time we use value iteration to learn the optimal value function for each state and goal pair, and encode it as Euclidean distance between each of those pairs. Then, planning can be done via a 1-step greedy policy on the representation space by comparing Euclidean distances of neighboring states to the goal and choosing the shortest.
>
> (3) How is the UVFA V used?
>
> V is used to learn a representation that is globally consistent. The V is equivalent to Euclidean distance in the learned representation space, and therefore greedy policy in the representation space gives the 1-step greedy policy obeying V.
>
> (4) When to use UVFA? When to use Dijkstra? Why are the choices in the experiments as made?
> UVFA is the main method of the paper, which learns a representation which can plan in O(1). Dijkstra’s is equivalent to optimal policy, which we use for analysis of optimal path lengths.

---

### Official Review · AnonReviewer2 · 2019-10-24
**Official Blind Review #2**

**Rating:** 3

**Review:**

## Paper Summary

While cast slightly differently in the intro, it seems to me that this paper learns a goal-conditioned value function that is used at test time to construct visual plans by selecting an appropriate sequence of data points from the training data. Similar to prior work they learn a local distance metric without supervision using a temporal proximity objective and then construct a graph of the training data points using this metric. The main novelty that this paper introduces seems to be the idea to distill the results of planning algorithms run at training time into a global, goal-conditioned value function, which allows to reduce the required planning time at test time. The authors perform experiments on constructing visual plans for a simulated toy navigation task, a robotic rope manipulation task and the StreetLearn navigation task. The paper reports favorable results under a time-constrained test setting but does not include strong baselines that were designed for this setting.

## Strengths

- bootstrapping a learned local distance metric to a global distance metric to reduce test-time planning cost is an interesting problem
- the paper has nice visualizations / analysis on the toy dataset
- the learning procedure for the local distance metric is clearly described
- the paper uses a large variety of different visualizations to make concepts and results clearer

## Weaknesses

(1) missing links to related work: the author's treatment of related work does not address the connections to some relevant papers (e.g. [1-3]) or is only done in the appendix (especially for [4]). It is not clearly delineated between techniques and ideas that are introduced in other papers (see (2) below) and the novel parts of this work. This makes it hard to understand the actual contributions of this paper.

(2) only minor contribution: the core parts of this paper build heavily on prior work: time-contrastive objectives for distance learning have been introduced in [5] and also been used in a very similar setup as here in [4], further [4, 3] also use semi-parametric, graph-like representations for planning with a learned local distance metric. The major contribution seems to be to distill the plans derived with either (a) n-step greedy rollout or (b) Dijkstra graph-search into a value-function so that planning does not need to be performed at test time. This somewhat small contribution is in contrast to the claims from the introduction that this paper "pose[s] the problem of unsupervised learning a plannable representation as learning a cognitive map of the domain".

(3) comparison to weak baselines: the main comparison in the experimental section is to a version of [4] where the authors constrain the planning horizon to a single step, which means effectively greedily using the local metric from Sec. 3.1. To be clear: this is in no way the method of [4]: they use Dijkstra-based planning at test time and it is clear that a "version" of [4] that does not use planning is not able to work. To me this seems rather like an ablation of the proposed method than a real baseline. The baseline that plans greedily with embeddings based on visual similarity has even less hope of working. The paper lacks thorough comparison to (a) baselines with the same semi-parametric structure that perform planning at test time (like the real method of [4]) and (b) methods that generate reactive policies without constructing a semi-parametric memory (e.g. off-policy RL). Only then a thorough comparison of pros and cons of planning at training/test time is possible (see detailed suggestions below).

(4) lack of qualitative samples for generated plans: for both the rope and the StreetLearn domain the authors do not provide thorough evaluation. For the rope domain only a single qualitative rollout is shown, for the StreetLearn domain no qualitative samples are provided for either the proposed method or the comparisons. (see suggestions for further evaluation below)

(5) explanation of core algorithmic part unclear: the explanation of how the local metric is used to learn the global value function is somewhat unclear and the used notation is confusing. Key problems seem to be the double-introduction of symbols for the local metric in Alg. 2 and the confusing usage of the terms "global embedding" and "value function" (see detailed questions below)

(6) terms used / writing structure makes paper hard to follow: the connection between used concepts like "global embedding", "plannable representation" and "goal-conditioned value function" are not clear in the writing of the paper. The authors talk about concepts without introducing them clearly before (e.g. problems of RL are listed in the intro without any prior reference to RL).

(7) lacks detail for reproducing results: the paper does not provide sufficient detail for reproducing the results. Neither in the main paper nor in the appendix do the authors provide details regarding architecture and used hyperparameters. It is unclear what policy was used to collect the training data, it is unclear how the baselines are working in detail (e.g. how the 1-step planning works) and how produced plans are checked for their validity.


## Questions

(A) What policy is used to collect the training data on each environment?
(B) What is the relation between the "global embedding" \Phi and the "goal-conditioned value function" V_\Phi(x, x_prime) in Algorithm 2?
(C) What is the difference between the local metric function \phi and the reward function in Algorithm 2? Are they the same?
(D) If they are the same, how can the local metric accurately estimate rewards for states x and x_g that are far apart from one another as would naturally be the case when training the value function?
(E) What does the notation N(1, \eps) in line 5 of Algorithm 2 mean?
(F) What is the expectation over the length of trajectories between start and goal on the StreetLearn environment (to estimate what percentage of that the success horizon of 50 steps is)?


## Suggestions to improve the paper

(for 1) please add a more thorough treatment of the closest related works on semi-parametric memory + learned visual planning + learned distance functions (some mentioned below [1-5]) to the main part of the paper, clearly stating differences and carving out which parts are similar and where actual novelty lies.

(for 2) please explain clearly the added value of distilling the training-plans into a value function for O(1) test-time planning and point out that this is the main difference e.g. to [4] and therefore the main contribution of the paper.

(for 3) in order to better understand the trade-offs between doing planning at test time (like [3,4]) or learning an O(1) planner contrast runtime and performance of both options (i.e. compare to the proper method of [4]). This will help readers understand how much speed they gain from the proposed method vs how much performance they loose. It might also make sense to include an off-policy RL algorithm (e.g. SAC) that uses the local metric as reward function (without constructing the graph) to investigate how much planning via graph-search can help at training time. Another interesting direction can be to investigate the generalization performance to a new environment (e.g. new street maze, new rope setup) after training on a variety of environment configurations. [3] showed that explicit test-time planning performs better than "pure" RL, it would be interesting how the proposed "hybrid" approach performs.

(for 4) please add randomly sampled qualitative results for both environments and all methods to the appendix. It can additionally be helpful to add GIFs of executions to a website. It might also be interesting to add a quantitative evaluation for the plans from the rope environment as was performed in Kurutach et al. 2018.

(for 5) please incorporate answers to questions (B-E) into the text in Sec 3.2 explaining Algorithm 2. It might also help to structure the text in such a way as to follow the flow of the algorithm.

(for 6) restructure and shorten the introduction, clarify terms like "inductive prior within image generation" or "non-local concepts of distances and direction" or "conceptual reward" or "planning network", clarify how the authors connect the proposed representation learning objective and RL. Avoid sentences that are a highly compressed summary of the paper but for which the reader lacks background, like in the intro: "training a planning agent to master an imagined “reaching game” on a graph".

(for 7) add details for architecture and hyperparameters to the appendix, add details for how baselines are constructed to the appendix. add details about data collection and evaluation for all datasets to the appendix (e.g. how is checked that a plan is coherent in StreetLearn). It might also help to add an algorithm box for the test time procedure for the proposed method.


## Minor Edit Suggestions
- Fig 2 seems to define the blue square as the target, the text next to it describes the blue square as the agent, please make coherent
- for Fig 7: the numbers contained in the figure are not explained in the caption, especially the numbers below the images are cryptic, please explain or omit


[Novelty]: minor
[technical novelty]: minor
[Experimental Design]: Okay
[potential impact]: minor

################
[overall recommendation]: weakReject - The exposition of the problem and treatment of related work are not sufficient, the actual novelty of the proposed paper is low and the lack of comparison to strong baselines push this paper below the bar for acceptance.
[Confidence]: High


[1] Cognitive Planning and Mapping, Gupta et al., 2017
[2] Universal Planning Networks, Srinivas et al., 2018
[3] Search on the Replay Buffer: Bridging Planning and Reinforcement Learning, Eysenbach et al., 2019
[4] Semi-Parametric Topological Memory for Navigation, Savinov et al., 2018
[5] Time-Contrastive Networks, Sermanet et al., 2017


### Post-rebuttal reply ###
I appreciate the author's reply, the experiments that were added during the rebuttal are definitely a good step forward. The authors added comparison to a model-free RL baseline as well as proper comparison to a multi-step planning version of SPTM. However, these comparisons were only performed on the most simple environment: the open room environment without any obstacle. These evaluations are not sufficient to prove the merit of the proposed method, especially given that it is sold as an alternative to planning methods. The method needs to be tested against fair baselines on more complicated environments; the current submission only contains baselines that *cannot* work on the more complicated tasks. I therefore don't see grounds to improve my rating.

**Experience Assessment:**

I have published one or two papers in this area.

**Review Assessment: Checking Correctness Of Derivations And Theory:**

I carefully checked the derivations and theory.

**Review Assessment: Checking Correctness Of Experiments:**

I carefully checked the experiments.

**Review Assessment: Thoroughness In Paper Reading:**

I read the paper thoroughly.

---

> ### Author Response · Authors · 2019-11-15
> **Thank you for your thoughtful review! We have incorporated these into the updated draft, with three new experiments.**
>
> Thank you for your thoughtful comment and constructive review, and this great opportunity to improve our paper.
>
> We incorporated all the suggested improvements, clarifications with respect to the method, and details to enable reproducibility in an updated draft. Specifically, we have included a DQN baseline which learned from a fixed dataset, but does not explicitly build the graph. Our new experiment shows that by constructing a graph, Plan2vec is able to attain success with much less data than off-line Q learning. Please refer to Fig. 6 in the updated draft.
>
> * We decided to use DQN instead of SAC here because the action space is discrete. SAC would work better if the action space is continuous. Note our state space is continuous, and the action space is discretized in the simulated navigation domain.
>
> We have also fixed the SPTM baseline to use the original, full method. We want to show a more complete picture, so in the updated draft, we added a new figure plotting the planning success rate of Plan2Vec, SPTM versus a random baseline, w.r.t. a range of planning budget. By increasing the plan-ahead horizon, both methods improve in success rate. But Plan2Vec gains a large gap due to its incorporation of an amortized value function as a planning heuristic, since the local simularity function that SPTM uses is limited to a small neighborhood.
>
> To give an idea of how much distance evaluation Dijkstra needs, we also added a figure comparing the empirical inference time between Plan2vec and Dijkstra. We have also included further discussion into the benefits of using the learned representation space to get O(|E|) planning time as opposed to O(|E|+|V|log|V|) which Dijkstra’s gets.
>
> We’ve also added an additional section in the Appendix detailing architecture and hyperparameter details for easier reproducibility.
>
> --------
>
> Detailed Responses for each specific comment:
>
> (A) What policy is used to collect the training data on each environment?
>
>      We use a random policy to collect the passive dataset for each environment.
>
> (B) What is the relation between the "global embedding" \Phi and the "goal-conditioned value function" V_\Phi(x, x_prime) in Algorithm 2?
>
> 	V_\phi(s, g):=||\phi(s) - \phi(g)||_2, defined as the Euclidean distance between the two states in the learned embedding space.
>
> (C) What is the difference between the local metric function \phi and the reward function in Algorithm 2? Are they the same?
>
>         The local metric function is used as a cost function for the MDP, so yes they are the same.
>
> (D) If they are the same, how can the local metric accurately estimate rewards for states x and x_g that are far apart from one another as would naturally be the case when training the value function?
>
> 	The cost is a constant negative factor with each step taken, it is not a shaped reward of distance between states x and x_g.
>
> (E) What does the notation N(1, ϵ) in line 5 of Algorithm 2 mean?
>
> 	We meant the ϵ neighborhood around 1, i.e. [1-ϵ, 1+ϵ], and updated the paper with the explicit definition.
>
> (F) At convergence, it takes roughly 8 planning steps. For each one of the planning steps, we search the neighborhood up to the 3rd neighbor. This corresponds to roughly 4-6 neighbors (2-3 on each side) in the Manhattan dataset. This is roughly 3 * 8 = 24 units, half of the 50 unit radius. On the Manhattan-tiny dataset, this is roughly 4 blocks in diameter.

---

### Official Review · AnonReviewer1 · 2019-10-29
**Official Blind Review #1**

**Rating:** 1

**Review:**

The paper aims at learning a latent representation of images in the setting of sequential decision making. The representation are learned such that there exists a metric that correctly reflects the difference in reachability between points in the neighborhood of the current observation and the goal.

I'm struggeling to understand the core approach of the paper. More specifically, while learning the local metric (Alg. 1) seems clear, I can not understand the details of Alg.2 (which btw. is never referenced in the text). The surrounding paragraph is not detailed enough. Why is \Phi(x, x') denoted a global embedding? \Phi has two inputs, shouldn't that be some sort of a metric? How is "find n" done? There is a lot of talk about embeddings, but they are actually not occuring in section 3. What is a 'plannable representation'?

Some of the experiments compare to VAE and its learned embedding space. Shouldn't the comparision be to models that think about the Riemannian geometry of a VAE, e.g. "Latent space oddity: on the curva-ture of deep generative models". There are a several citations missing in that direction, going back to at least "Metrics  for  probabilistic geometries" by Tossi et al. in 2014. As it was also pointed out in a public comment, relevant citations related to "Learning for planning" seem to be missing, too. Finally, a wider set of experiments needs to demonstrate the method (again, considering the scope of the as-of-now-not-cited papers).

**Experience Assessment:**

I have read many papers in this area.

**Review Assessment: Checking Correctness Of Derivations And Theory:**

I assessed the sensibility of the derivations and theory.

**Review Assessment: Checking Correctness Of Experiments:**

I assessed the sensibility of the experiments.

**Review Assessment: Thoroughness In Paper Reading:**

I read the paper at least twice and used my best judgement in assessing the paper.

---

> ### Author Response · Authors · 2019-11-15
> **Response to R1**
>
> Thank you for your thoughtful comments! We have completely re-written the method section of the paper, and have added three more experiments showing the improved sample complexity of our method compared with standard off policy method, and the improved planning performance of plan2vec versus SPTM under a range of planning budget. Please refer to the revised draft for the additional references, and the updated website for more qualitative results.
>
> Detailed responses:
>
> - We apologize for the Methods section being unclear! Please refer to the updated draft for the revised version.
>
> - It is indeed a mistake in Alg 2 for $\Phi$ to take in $x$ and $x’$ and be called an embedding function.
>
> To clarify the notations, we decided to refer to all metric functions as $f_\phi$, where the subscript $\phi$ indicates the embedding we use. Both $\phi$ and $\Phi$ map states to a representation space. The value function is constructed by either taking the $L_p$ distance between the two latent vectors, or by passing them through a multi-layer perceptron. Plan2vec treats the local metric function $f_\phi(x, x')$ as the negative reward, and the global metric function $f_\Phi(x, x')$ as the negative value function. By running value iteration using the plans sampled on the graph, using this global metric function as a planning heuristic, plan2vec bootstraps the information in the local metric function to the global metric function. In doing so also obtaining a compact representation in the form of the global embedding $\Phi$.
>
> We have experiments exploring different kernels and different type of (pseudo-) metric that one can impose on the global embedding, that we plan to introduce to later versions of the paper.
>
> - “Find n” is done using the local metric, which outputs a continuous score denoting closeness in time steps. We can find all 1-step neighbors by passing in all pairs of states to f_\phi, and taking those that are in neighborhood [1-\epsilon, 1+\epsilon].
>
> - We have incorporated more of the relevant citations relating to learning for planning, as well as incorporating experiments comparing against the full SPTM setting, showing computational trade-offs with that method, and with Soft Actor-Critic, as a 1-step greedy policy method.
>
> We appreciate the references to additional generative models. The VAE visualizations intend to show how generative models do not learn representations that extend well to sequential data and planning. The referenced works do not extend to sequential data, which is the setting we are studying.

---

### Public Comment · ~Aravind_Srinivas1 · 2019-10-12
**Insufficient coverage of related work and re-introduction of existing ideas**

I find it quite necessary to point out the following facts:

1. The paper tries to introduce plannable representation learning as a problem they study and contribute to without doing justice to some of the better papers on this topic such as
a. Value Iteration Networks - Tamar et al 2016
b. Cognitive Mapping and Planning - Gupta et al 2017
c. Universal Planning Networks - Srinivas et al 2018
d. Gated Path Planning Networks - Lee et al 2018
e. Distributional Planning Networks - Yu et al 2019

2. InfoNCE / NCE for learning distance metrics - a. Time Contrastive Networks - Sermanet et al 2018, b. Warde-Farley et al 2019 - have already done the same idea (arguably better versions) and not been cited.

3. In fact, the notion of plannable representations (what does that even mean and why is it worth a problem studying, what does it mean to generalize to unseen tasks through plan-based priors, why generative approaches are ill-suited for this problem, why building effective abstract maps/distance-aware representations of the raw observation space is useful, how distance metrics can be used for planning through dense smooth rewards, etc have been extensively discussed in Tamar et al and Srinivas et al).  The idea of building a cognitive visual map for path planning is the main contribution in Gupta et al.

4. Finally, way more impressive results have been shown on more complex tasks than those considered in this paper (simple maze navigation and uncluttered rope images) in prior work.

---

> ### Author Response · Authors · 2019-11-15
> **Addressing Related Work**
>
>
>
> Hi Aravind,
>
> Thank you for reading our paper! We have updated the draft to include these related works, and moved the related work section to the main text (it was placed in the appendix due to space constraints, a decision the authors regret). We will post it here as soon as it is ready for upload.
>
> In comparison with VIN, GPPN and CMP [3-5], a major difference is that Plan2vec does not assume that the feature vectors live on a 2D grid world with known environment dynamics and well-defined local connectivity between states. Instead, Plan2vec is more akin to graph-based approaches such as SPTM, DeepWalk and diffusion maps. Plan2vec learns the local connectivity of the domain contrastively, and runs the value iteration through graph-convolution. This is numerically more difficult as reported in [5], and according to our experience. Also note the number of neighbors for each node is not a constant, but is conditioned on each sample.
>
> Both UPN and DPN [1, 2] require expert observations paired with action data. They rely on supervised learning to get to the reward through a differentiable forward model, trained end-to-end [0] by grounding through expert actions. Plan2vec in this regard follows Causal InfoGAN, where the dataset (rope) only contains observations but not the actions. This is a common scenario with human demonstrations and deformable object manipulation tasks because actions are not explicitly observable. To get around the requirement of having an expert sampling policy, Plan2vec constructs a graph from the dataset, which allows it to sample segments from multiple different trajectories. This alleviates the need for each trajectory to be optimal in its entirety.  Additionally, Plan2vec can also generalize to longer planning horizons beyond the sampled trajectories used during learning, as shown in the rope experiments; UPN and DPN on the other hand only guarantee generalization to trajectory lengths observed during training.
>
> Another contribution of our work is to demonstrate that with unsupervised representation learning, we can go beyond conditionally generating samples in a small temporal window, and learn features that are not visually apparent. We demonstrate the first point by showing that our method can learn a plannable representation on the rope domain without image generation. Our result is superior to ones that appeared in [7] because the planning horizon is much longer (hundreds of steps vs just a few). The simulated rope domain in [2] has a similar difficulty level because the background is fixed for the entire simulated dataset, making it trivial for the convnet to learn to ignore [8].
>
> We demonstrate the second point with results on the StreetLearn dataset, a complex real world dataset where the location is hard to identify from individual streetview alone. We provide quantitative results, and show that Plan2vec achieves good performance on this domain in comparison to baselines. Additionally, it generalizes to initial and goal state pairs that it has not seen during training.
>
>
> [0] An On-line Algorithm for Dynamic RL and Planning, Schmidthuber et al
> [1] Universal Planning Networks, Srinivas et al
> [2] Unsupervised Visuomotor Control through Distributional Planning Networks, Yu et al
> [3] Value Iteration Networks, Tamar et al
> [4] Gated Path Planning Networks, Lee et al
> [5] Cognitive mapping and planning for visual navigation, Gupta
> [6] From Language to Goal, Fu et al
> [7] Learning plannable representations with Causal InfoGAN, Kurutach
> [8] Learning Robotic Manipulation through Visual Planning and Acting, Wang et al

---

### Author Response · Authors · 2019-11-15
**Overall response**

We thank the reviewers for detailed comments and constructive suggestions to improve the paper.

Since the review period, we re-wrote large sections of the draft. We have added missing citations from the control and RL side, clarified the method, added hyperparameters and implementation details, and relevant qualitative results on our website.

In response to the constructive reviews, we also added a number of new experiments, to highlight the benefit of combining the idea of 1. constructing a graph on top of a passive dataset, and 2. extrapolating a local metric function to an amortized value function, for planning.

In this updated draft, we incorporate:

1. an additional baseline of DQN
2. a full SPTM baseline over a range of different planning budget.
3. Additional analyses of the computation at inference time of our method in comparison to Dijkstra’s.
4. We have also included empirical results of computation needed for various planning lengths of the two methods.

For these new experiments, please refer to Fig. 6 in the updated paper.

For improved clarity, we:

1. clarified the introduction
2. added missing definitions for terms
3. re-written the methods section to explain how the local metric is used to generate neighbors in place of a dynamics model, how UVFA works in the learned representation space.

We hope the reviewers find the new draft of the paper to be much clearer. Please refer to the personalized responses below for detailed responses

---

### Decision · Program_Chairs · 2019-12-19

**Decision:**

Reject

**Comment:**

The paper proposes a representation learning objective that makes it
amenable to planning,

The initial submission contained clear holes, such as missing related work and only containing very simplistic baselines. The authors have substantially updated the paper based on this feedback, resulting in a clear improvement.

Nevertheless, while the new version is a good step in the right direction, there is some additional work needed to fully address the reviewers' complaints. For example, the improved baselines are only evaluated in the most simple domain, while the more complex domains still only contain simplistic baselines that are destined to fail. There are also some unaddressed questions regarding the correctness of Eq. 4. Finally, the substantial rewrites have given the paper a less-than-polished feel.

In short, while the work is interesting, it still needs a few iterations before it's ready for publication.